# Evaluation of Long–Lasting Antibacterial Properties and Cytotoxic Behavior of Functionalized Silver-Nanocellulose Composite

**DOI:** 10.3390/ma14154198

**Published:** 2021-07-27

**Authors:** Roberta Grazia Toro, Abeer Mohamed Adel, Tilde de Caro, Fulvio Federici, Luciana Cerri, Eleonora Bolli, Alessio Mezzi, Marianna Barbalinardo, Denis Gentili, Massimiliano Cavallini, Mona Tawfik Al-Shemy, Roberta Montanari, Daniela Caschera

**Affiliations:** 1Institute for the Study of Nanostructured Materials, National Council of Research, Via Salaria km 29,300, Monterotondo, 00015 Rome, Italy; robertagrazia.toro@cnr.it (R.G.T.); tilde.decaro@cnr.it (T.d.C.); fulvio.federici@cnr.it (F.F.); luciana.cerri@cnr.it (L.C.); eleonora.bolli@ismn.cnr.it (E.B.); alessio.mezzi@cnr.it (A.M.); 2Cellulose and Paper Department, National Research Centre, 33El-Bohouth St. (Former El-Tahrir St.), Dokki, Giza, Cairo 12622, Egypt; abeermadel2003@yahoo.com (A.M.A.); mt.el-shemy@nrc.sci.eg (M.T.A.-S.); 3Institute for the Study of Nanostructured Materials, National Council of Research, Via P. Gobetti, 40129 Bologna, Italy; marianna.barbalinardo@ismn.cnr.it (M.B.); denis.gentili@cnr.it (D.G.); massimiliano.cavallini@cnr.it (M.C.); 4Institute of Crystallography, National Council of Research, Via Salaria Km 29,300, Monterotondo, 00015 Rome, Italy; roberta.montanari@cnr.it

**Keywords:** green synthesis, silver-cellulose nanocomposites, long lasting antibacterial properties, cytotoxicity

## Abstract

Materials possessing long-term antibacterial behavior and high cytotoxicity are of extreme interest in several applications, from biomedical devices to food packaging. Furthermore, for the safeguard of the human health and the environment, it is also stringent keeping in mind the need to gather good functional performances with the development of ecofriendly materials and processes. In this study, we propose a green fabrication method for the synthesis of silver nanoparticles supported on oxidized nanocellulose (ONCs), acting as both template and reducing agent. The complete structural and morphological characterization shows that well-dispersed and crystalline Ag nanoparticles of about 10–20 nm were obtained in the cellulose matrix. The antibacterial properties of Ag-nanocomposites (Ag–ONCs) were evaluated through specific Agar diffusion tests against *E. coli* bacteria, and the results clearly demonstrate that Ag–ONCs possess high long-lasting antibacterial behavior, retained up to 85% growth bacteria inhibition, even after 30 days of incubation. Finally, cell viability assays reveal that Ag-ONCs show a significant cytotoxicity in mouse embryonic fibroblasts.

## 1. Introduction

Several deadly infections originated by bacteria and viruses can seriously threat human health [1]. Thus, the development of materials and devices with long lasting antibacterial effects and high cytotoxic behavior is of great importance [2]. Recently, cellulose and its derivatives have found several applications in the fabrication of environmentally friendly and biocompatible products due to their excellent physical and biological properties, biocompatibility and biodegradability [3]. Cellulose itself usually does not show antibacterial properties, and its extreme hygroscopicity can concur to the creation of a breeding ground for bacteria growth [4], limiting strongly the application of cellulose-based materials in technological fields such as biomedicine, cosmetics, food packaging, and tissue engineering [5]. However, the opportune combination of cellulose with organic and/or inorganic materials can produce an effective biomedical agent with antibacterial activity. Among the inorganic materials, significant results were recently achieved through the functionalization of several metal and metal oxide nanoparticles such as Au [6], Cu [7], TiO_2_ [8], and ZnO [9] because of their chemical stability, robustness, and efficient bactericidal action. In particular, silver was known as effective bactericidal agent in food and water preservation since ancient Egyptians and Romans [10]. Nowadays, Ag nanoparticles are widely used in the medical field for a variety of applications [11,12], even in those cases in which different antibiotic free strategies would be applied [13,14], for the low-toxicity to human cells [15]. Silver is also intensively studied as an effective anticancer agent for the clinical management of cancer, since the oxidative stress mechanism induced by its cytotoxic effect may be able to reduce the viability and modify the morphology of cancer cells [16,17].

Furthermore, silver nanoparticles were used in conjugation with biomolecules, such as cellulose and chitosan, to obtain organic-inorganic composite with specific antibacterial activity [18,19,20] and resistance against bacteria, fungi, protozoa, and virus contamination [11,21,22,23]. Nevertheless, the over-using of extra toxic reducing reagents or capping/dispersing agents during the synthesis of uniform and stable cellulose-Ag nanohybrids strongly limit their application in health sector [24] and represent a cause of environmental pollution [25].

The antibacterial effect should have a discrete durability to avoid a rapid degradation of the materials due to the growth of the bacteria on the surfaces. In this contest, the development of greener synthetic methods for the preparation of Ag-based nanocomposites with specific biological activities still needs to be explored. Furthermore, the environmental impact of silver nanoparticles is still a controversial issue [26], as they can transform in the environment due to interaction with some components, which can modify their final toxic effects. For example, Ag NPs contained in sock fabrics can easily end up into waste water during the washing cycles [27], reducing the activity of microbial-activated sludge systems in waste water treatment. At the same time, when they are accidentally release in the soil, they can affect the beneficial activities of certain bacteria in the soil, which are essential for agricultural crops [28]. However, opportune strategies should be considered to obtain silver-based nanocomposites not easily dispersible in the environment and suitable for strong immobilization in porous substrates (paper, textile, sponge), avoiding further dispersion of the active materials.

Here, we report a facile, green, and highly efficient chemical route for the fabrication of Ag NPs-nanocellulose nanocomposites (Ag–ONCs). The structure and chemical composition of the Ag–ONCs nanocomposite were investigated by X-ray Diffraction (XRD), Fourier Transformed—Infra Red (FT-IR) and Raman spectroscopy, and X-ray Photoelectron Spectroscopy (XPS) techniques. Our results highlight that under UV photo-irradiation, the aldehydic and carboxylic groups of the cellulose serve as reducing agents and stabilizers to the formation of well-dispersed silver nanoparticles within the cellulosic matrix. The distribution of Ag nanoparticles in the cellulosic matrix and their homogeneity in size were evaluated by a complete morphological analysis by scanning electron microscopy (SEM) atomic force microscopy (AFM), and transmission electron microscopy (TEM) techniques. Ag–ONCs already demonstrated very good antibacterial properties, being able to inhibit the growth of both gram positive and gram-negative bacteria, with a growth inhibition up to 99% against *E. coli* [18]. Nevertheless, for practical applications in food packaging or medical devices, it is extremely important that the antibacterial activity exhibited by the materials remain high for a longer time and in less favorable ambient conditions. This should exclude the rapid degradation of the materials due to the bacterial growth on the surfaces. Our investigation shows that the long–lasting protection of Ag–ONCs vs. *E. coli* growth is effective for at least one month, even after storing the system in the dark to avoid the effect of silver activation by light. Finally, to pave the way for a possible application in biomedicine, the cytotoxic properties of Ag–ONCs nanocomposites in mouse embryonic fibroblasts (NIH-3T3) were assessed as well and the results confirm that the observed toxic activity of the Ag–ONCs nanocomposites can be attributed to silver.

## 2. Materials and Methods

All chemicals are of are purchased by Sigma–Aldrich (St. Louis, MO, USA), analytical grade. The bagasse fibers are kindly supplied from Quena Paper Industrial Company (Kous Mill, Kous, Egypt).

Ag-ONCs nanocomposite preparation: ONCs was prepared from bagasse raw material using ammonium persulphate hydrolysis method as reported elsewhere [29] and used without any other purification. The Ag–ONCs composite are synthesized by adding drop-wise an AgNO_3_ solution (0.02 M) to a ONCs water suspension. The amount of AgNO_3_ precursor is calculated to obtain a final weight ratio Ag/ONCs of 20:80 wt.% The starting solution was stirred for 15 min, then put under UV lamp illumination (Spectroline^®^ E-Series lamp bulb, Model EN-160, 6 W, λ = 354 nm) for 300 min. The success of the silver reduction is asserted by using a great excess of reducing/stabilizing agent (carboxylated nanocellulose) for an irradiation time (300 min) long enough to consider the reaction complete and quantitative (the UV/Vis spectrum did not change anymore after 240 min of UV irradiation). The change of the solution color from transparent to brownish is also observed (as illustrated in Figure 1a, inset). However, we cannot exclude that very small amounts of Ag+ (not revealed by the instrumental resolutions) could be yet in the colloidal solution. The Ag–ONCs suspension results to be stable for more than 12 months, with no deposition or flocculation detected in the overall period.

### 2.1. Structural and Morphological Characterization

Structural characterizations of ONCs and Ag–ONCs are carried out by UV-Vis spectroscopy, X-ray diffraction (XRD), Attenuated Total Reflectance Fourier Transform Infrared (ATR-FTIR), micro-Raman spectroscopy and X-ray Photoelectron spectroscopy (XPS). Few drops of the ONCs and Ag–ONCs solutions were deposited on a Si substrate and dried at room temperature in air. In some cases, due to the low thickness of the deposited samples layer, some peaks related to the silicon substrate could be observed.

UV-vis absorbance spectra are collected in the wavelength range 300–870 nm using a double beam spectrophotometer V-660 (Jasco, Tokio, Japan), using a quartz cell of thickness 1 cm.

XRD are recorded on powder samples in the range 2θ = 5–80°, by a Siemen D5000 X-ray diffractometer (Boston, MA, USA), using Cu kα (λ = 1.5406°A) radiation and operating at 40 kV and 30 mA.

ATR-FTIR spectra are recorded on a Shimadzu IRPrestige-21, using an ATR-8200HA accessory, with a ZnSe crystal tip, in the frequency range 4000–500 cm^−1^, at a resolution of 4 cm^−1^ and 200 scans.

µ-Raman analysis is recorded at room temperature with a Renishaw RM 2000 (Gloucestershire, UK), using an Ar^+^ laser (514.5 nm excitation line), equipped with a Peltier cooled charge-coupled device (CCD) camera and a 50X objective Leica optical microscope. The laser power was optimized to avoid the silver oxidation laser-induced.

XPS measurements are carried out using an ESCALAB MkII spectrometer (Thermo Fisher, MA, USA) equipped with a nonmonochromatized Al Kα source and five channeltrons as detection system. The analyzed sample is prepared drying, at ambient conditions, a drop of nanocomposite solution on a Si-SiO_2_ substrate. The spectra are collected operating at constant pass energy of 40 eV, at a base pressure of about 10^−7^ Pa and the binding energy scale is calibrated respect with the the C 1s peak f at BE = 285.0 eV. All data are collected and processed by Avantage V.5 Software.

Morphological characterization are examined through a Dimension 3100 Atomic Force Microscope (Bruker, MA, USA) equipped with a Nano Scope IIIa, controller (Veeco, Santa Barbara, CA, USA) operating in tapping mode and by a high resolution TEM with a JEOL JEM-2100 (Tokyo, Japan), while SEM observations are carried out using a Leo 1530 microscope (ZEISS, Oberkochen, Germany), equipped with an Oxford 30 mm^2^ SDD EDS and a KE Developments CENTAURUS detector, using the same parameter for both ONCs and Ag–ONCs sample (magnification 150.00 K, EHT = 5 kV). ONCs and the Ag–ONCs are deposited from an aqueous 10 times diluted dispersion onto silicon substrate for AFM and SEM and on a micro grid covered with a thin carbon film (≈200 nm) for TEM and let dry on air at room temperature. For TEM measurements, the deposited ONCs are stained with a 2% uranyl acetate solution to enhance the microscopic resolution, while deposited Ag-ONCs left without staining.

### 2.2. Antibacterial Tests

The antibacterial activity of ONCs and Ag-ONCs nanocomposites are evaluated using Agar diffusion method. *Escherichia Coli* was used as gram negative bacteria and agar petri dishes are freshly prepared. Before spreading the bacterial suspension on the plates, the Ag–ONCs solution, containing 220 µg/mL of Ag, is spread on the test well and the solution containing only ONCs (880 µg/mL) is spread on the control well. The samples are incubated 15 min at 4 °C, then 200 μL of the bacterial suspension with OD_590_ = 0.6 diluted 10^−15^ in to LB broth are plated out into the wells. The samples are incubated at 37 °C and the growth of the bacteria was evaluated at different times. The antibacterial rate is calculated by Equation (1):C (%) = [(A − B)/A] × 100%(1)
where C represents antibacterial rates, A is the average number of colonies formed units in ONCs petri, and B is the average number of colonies formed units in Ag–ONCs samples.

### 2.3. Cell Cultures

Mouse embryonic fibroblast (NIH-3T3) cells are cultured in DMEM medium supplemented with: heat-inactivated FBS (10% *v/v*), L-glutamine (2 mM), MEM Non Essential Amino Acids (0.1 mM), penicillin (100 U/mL) and streptomycin (100 U/mL) in a humidified incubator (37 °C, 5% CO_2_) [30]. Cells are seeded in 96-well plates (5 × 10^4^ cells per mL) and grown for 24 h before exposure to ONCs and Ag-ONCs. 

Samples are examined using an optical inverted microscope (Eclipse TS100, NIKON, Tokyo, Japan).

### 2.4. Cell Viability (Cytotoxicity Assay)

Cell viability is assessed by 3-(4,5-di-methyl-2-thiazolyl)-2,5-diphenyltetrazolium bromide (MTT) assay [31]. Briefly, cells are seeded in 96-well plates with or without ONCs and Ag–ONCs for 24 h. After incubation times, the medium is discarded and cells are washed with DPBS. Afterwards, sterile MTT solution (110 µL, 0.45 mg/mL in DPBS) is added to each well and incubated for 1 h (37 °C, 5% CO_2_). Subsequently, the medium is discarded and DMSO (200 µL) is added to each sample to solubilize formazan crystals. Absorbance at 550 nm as a working wavelength and 640 nm as a reference are read using a microplate reader (Thermo Scientific, MA, USA, Varioskan Flash Multimode Reader). Cell viability is calculated as the proportion of the absorbance of cells treated with nanoparticles relative to that of the untreated cells (control).

## 3. Results and Discussion

### 3.1. Structural and Morphological Characterization of Ag-ONCs Nanocomposite

The success in the fabrication of Ag–ONCs nanocomposites was asserted through structural and morphological characterization techniques.

In Figure 1a, the UV-Vis spectra of the AgNO3 and nanocellulose solution before and after the irradiation at 354 nm for 300 min are reported.

Before the UV irradiation, starting solution is uncolored and no absorption is visible in the corresponding UV-Vis spectrum. After 300 min of UV irradiation at λ = 354 nm, the color of the suspension changes to red-brownish, and the UV-vis spectrum shows a very intense and broad absorption at about 450 nm, which is specific for Ag NPs. This UV–Vis absorption band represents the typical surface plasmon resonance (SPR) Ag nanoparticles. The observed red-shift for the Ag plasmonic maximum was attributed to the capping effect of the cellulose [32,33].

The formation of well crystalline Ag NPs in the nanocomposite is confirmed by XRD and the corresponding X-ray diffraction patterns of ONCs and Ag–ONCs are shown in Figure 1b. The XRD pattern of the Ag-ONCs nanocomposite displays the characteristic reflections of cellulose at 15.53°, 22.30°, and 34.71° corresponding to the (101), (200) and (400) lattice planes, respectively [34]. In addition, the diffraction reflections at 38.22° 44.77°, 64.43°, and 77.44°, attributed to the (111), (200), (220), and (311) crystal planes of face centered cubic (FCC) structure of metallic Ag NPs (JCPDS card no. 65-2871) are also present. The presence of reflections at 2θ = 38.22° and 44.77° for silver is usually associated with the formation of spherical or quasispherical nanoparticles [35]. No XRD peaks attributed to the silver nitrate precursor are detectable, assuming that the reduction of Ag^+^ in Ag^0^ could be considered complete.

The ATR-FTIR spectra of nanocellulose and nanocomposite are shown in Figure 1c. The FTIR of ONCs shows the typical bands of cellulose: the stretching and bending vibrations of OH groups at 3334 cm^−1^ and 1630 cm^−1^, the stretching vibrations of C–O–C at 1160 cm^−1^ and 1060 cm^−1^, the C–H stretching vibrations at 2890 cm^−1^, C–H bending deformations at 604 cm^−1^, and the wagging and deformation of the C-H vibration in the region between 1450 cm^−1^ and 1300 cm^−1^. The band of the C–OOH stretching at 1715 cm^−1^ is present as a small shoulder [36]. Compared to ONCs, the FTIR spectrum of Ag-ONCs nanocomposite exhibits a general reduction of the intensity of the bands related to OH vibrations, and a slight shift of the OH stretching band toward lower wavenumbers, which imply the important role played by the OH groups in the Ag NPs formation and stabilization [37]. Furthermore, the peaks between 1100 cm^−1^ and 1020 cm^−1^ become stronger, as the effects of an increase in the oxidation degree of the cellulose, thereby confirming the role of carboxyl functional groups in their reductant role for Ag NPs formation [38].

Raman investigation (as illustrated in Figure 1d) confirms the formation of Ag NPs in the cellulose matrix: the Raman spectrum of Ag–ONCs nanocomposite reports the peaks characteristic of the Ag lattice vibrational modes at 106 cm^−1^ and 244 cm^−1^ [39], in addition to the Raman peaks of cellulose [40]. The peak at 500 cm^−1^ arises from the silicon substrate on which the Ag–ONCs composite solution is spread for Raman measurement.

The surface chemical composition of the sample is investigated by XPS. In Table 1, the XPS quantification and the binding energy (BE) of the main peaks of the detected elements are listed. The cellulose fingerprint is identified by the investigation of C 1s and O 1s spectra, shown in Figure 2. As typical for cellulose, the C 1s signal is characterized by the presence of three components positioned at BE = 285.0 eV, 286.7 eV, and 288.5 eV, assigned to C–C and C–H bonds, C–O bond and C=O and O–C–O bonds, respectively [41]. Whilst the O 1s signal is characterized by two components, located at BE = 531.8 eV and 533.1 eV, and assigned to C–O and water, CO, O–C–O respectively. Small amounts of Na^+^ and sulphite are also detected. Their presence is attributed to the ONCs production method using agriculture bagasse waste.

The Ag 3d signal is characterized by the typical Ag 3d_5/2_–Ag 3d_3/2_ doublet separated in energy by 6 eV. The Ag 3d_5/2_ peak is positioned at BE = 368.2 eV, which could be assigned to the presence of metallic silver. To better identify the silver oxidation state, the Auger parameter α′ was calculated given by the sum of the binding energy of the core level peak (Ag 3d_5/2_) and the kinetic energy of the photoinduced Auger signal (Ag MNN). The obtained value α′ = 725.9 eV, indicates the presence of Ag only in its metallic oxidation state [42]. Thus, the XPS measurements testifies the presence of Ag NPs in the cellulosic structure of the Ag-ONCs [43].

The morphology of the ONCs and Ag-ONCs is investigated by TEM, SEM, and AFM (as illustrated in Figure 3a–f). Highly uniform rod shape ONCs particles of about 40–95 nm in length and 5.94–10.42 nm in width are clearly observable from TEM (as illustrated in Figure 3a). SEM and AFM (as illustrated in Figure 3b,c, respectively) confirm the good dispersion of the nanocellulose rods.

The images of Ag-ONCs are shown in Figure 3d–f, as TEM, SEM, and AFM analysis results, respectively. The morphological characterization indicates the formation of spherical nanoparticles of about 10 nm in diameter. The SEM micrograph of the Ag-ONCs nanocomposite (as illustrated in Figure 3e) shows the homogeneous dispersion Ag NPs within the cellulosic matrix, thanks to the ability of the carboxyl and hydroxyl groups of the ONCs to coordinate metal nanoparticles. Such strong interactions decrease the mobility of metal nanosystems, stabilizing them and preventing the growth of large particles.

### 3.2. Antibacterial Measurements

Agar diffusion method experiments are carried out to evaluate the effective antibacterial properties of the Ag–ONCs nanocomposites against *E. coli* after different incubation times (as illustrated in Figure 4).

The results clearly indicate that the Ag–ONCs nanocomposite has an effective antibacterial activity against *Escherichia coli* with respect to the ONCs control sample, inhibiting the growth of the bacteria colonies up to 72 h of incubation. The calculated antibacterial rate, C%, is estimated to be over 95% for the Ag–ONCs nanocomposite after 72 h incubation time. The effective antibacterial activity against Gram-negative bacteria suggests the suitability of Ag–ONCs nanocomposites in those applications that require long-term bactericidal performances. The long-lasting protective behavior of Ag–ONCs nanocomposite towards *E. coli* bacteria growth was further tested by carrying out two separate Agar diffusion experiments in which two sets of ONCs and Ag–ONCs were prepared and then stored separately: one set was kept in the dark, while the other was left under natural light illumination. In this way, it is possible to evaluate the influence of the UV component of the natural light on the effective Ag–ONCs antibacterial properties, since the light may enhance the Ag NPs toxicity, increasing the Ag particles release and their bacterial uptake [44,45].

Figure 5 and Figure 6 show the *E. coli* growth in ONCs and Ag–ONCs for the samples left under natural light and stored in the dark, respectively.

As expected, Ag–ONCs nanocomposites are active even when they are kept in the dark for long periods (30 days), reaching an antibacterial rate C% around 65%, a value comparable to those reported for similar systems [46]. Moreover, the best results were obtained for the sample stored under natural light: after 30 days of incubation, indeed, Ag–ONCs nanocomposite exhibits an antibacterial rate C% higher than 85%.

The antibacterial mechanism of the Ag–ONCs nanocomposites was discussed in our previous works [18,20], in which the antibacterial behaviour of the Ag–ONCs was attributed to the synergic effect of different functionalities on the cellulosic surface, combined with the intrinsic properties of Ag. Ag NPs seems to be able to interact with the cell surface of different bacteria, according to a mechanism already reported in other studies [12,47]. The high viscosity of the suspension and its low water solubility permit a controlled release of Ag NPs that can interact with sulfur-containing proteins in the cell membrane. In this way, Ag–ONC bionanocomposites can thus alter their permeability, lead to cell leakage, interfere and destroy the normal metabolism of cells, and infiltrate their inner shell [48].

### 3.3. Cell Viability

Toxicity studies are performed by exposure of mouse embryonic fibroblasts (NIH-3T3) to Ag–ONCs and ONCs and their cytotoxic activity is assessed by MTT assay. Since Ag NPs were reported to exhibit a dose- and coating-dependent toxicity in NIH-3T3 cells [49,50], we assessed a concentration of 30 μg/mL in silver of Ag–ONCs and the corresponding concentration of ONCs. As shown in Figure 7a, after 24 h of exposure, the MTT viability assay shows a drastic decrease in mitochondrial function of fibroblasts exposed to Ag–ONCs, while ONCs does not show a significant cytotoxic effect. In agreement with the MTT assay, optical images taken further confirm that Ag–ONCs significantly affect the growth of NIH-3T3 cells, revealing detachment of the cell layer (as illustrated in Figure 7b). By contrast, cells treated with ONCs (as illustrated in Figure 7c), similarly to untreated cells (as illustrated in Figure 7d), show no abnormalities, further confirming that the toxic activity of Ag–ONCs is due to the silver.

## 4. Conclusions

Here, we present a facile and green approach to fabricate a silver-based nanocomposite using oxidized cellulose nanocrystals as a template/reducing agent and UV light irradiation as radical initiator for the silver reduction. Structural and morphological characterizations of Ag–ONCs nanocomposite point to the success of the synthetic procedure and the results confirm the formation of highly crystalline, semispherical Ag nanoparticles of about 10–20 nm, homogeneously dispersed in the ONCs matrix. XRD, Raman, FT-IR, and XPS analyses agree with the presence of metallic silver and no silver precursor is detected in the colloidal solution within the limits of the experimental resolutions. Small amount of Ag+, eventually present in the suspension, cannot be ruled out.

Furthermore, the green-synthesized Ag–ONCs nanocomposites have strong antibacterial activity against *E. coli* by inhibiting bacterial growth, and their bactericidal effects remain high even when the system is stored in the dark. Ag–ONCs nanocomposites also show significant in vitro cytotoxicity against a cell line (embryonic fibroblasts NIH-3T3). These results clearly indicate that Ag–ONCs possess very good long lasting antibacterial activity against *E. coli* and good cytotoxicity, making them suitable candidates for such applications in which these peculiar properties are needed. 

## Figures and Tables

**Figure 1 materials-14-04198-f001:**
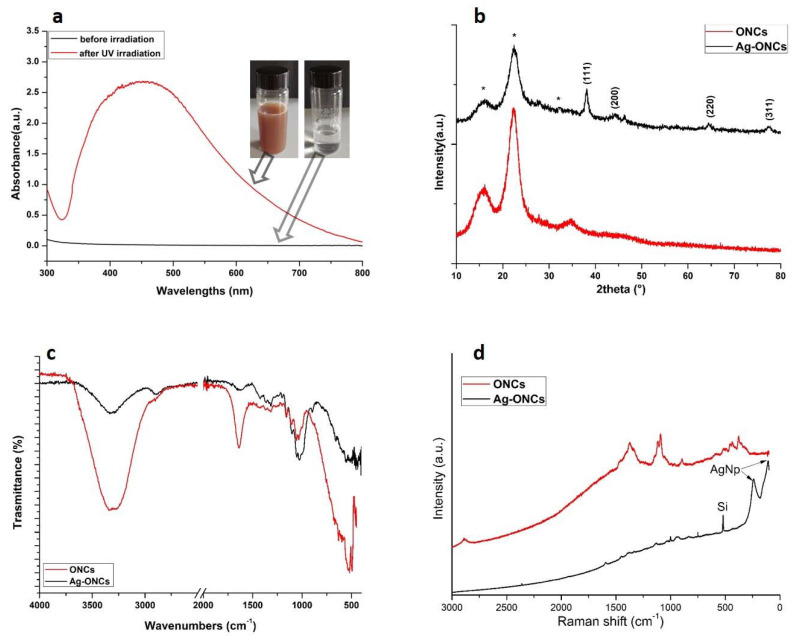
Comparison between for ONCs and Ag–ONCs: (**a**) UV–Vis before and after UV irradiation (**b**) X-ray diffraction (XRD), (**c**) Attenuated Total Reflectance Fourier Transform Infrared (ATR–FTIR), and (**d**) Raman spectra. Note that the * in (**b**) are referred to the XRD cellulose peaks in the nanocomposite.

**Figure 2 materials-14-04198-f002:**
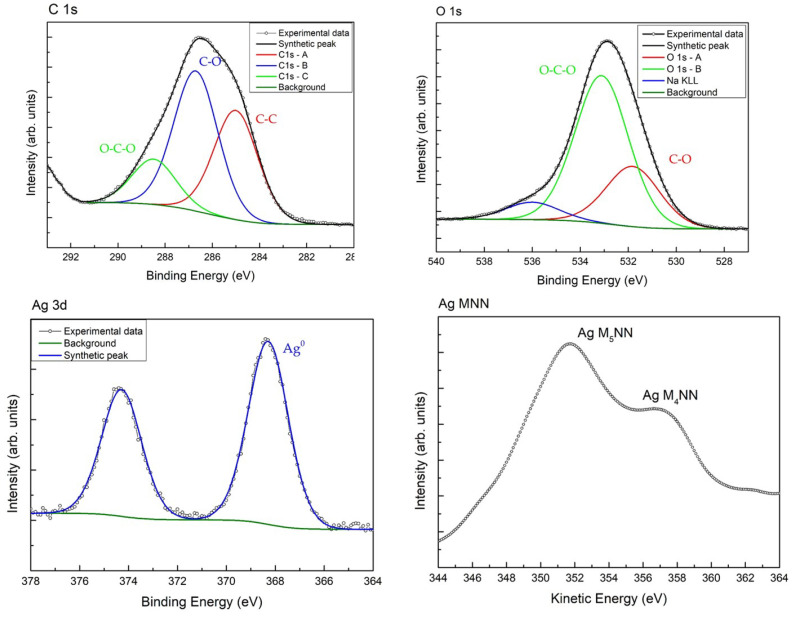
XPS spectra deconvolution for C 1s, O 1s, Ag 3d and Auger spectrum for Ag.

**Figure 3 materials-14-04198-f003:**
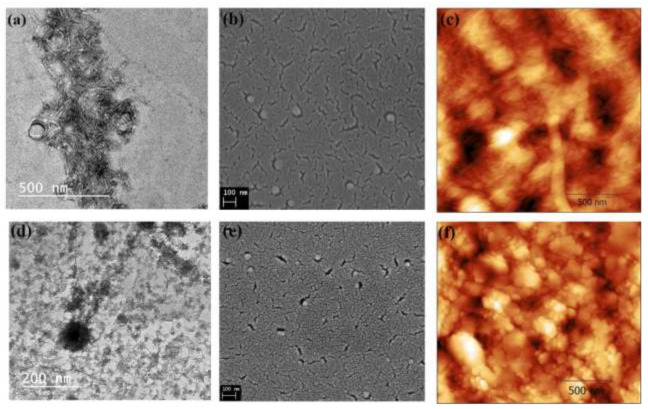
Morphological analysis for ONCs with (**a**) TEM, (**b**) SEM, and (**c**) AFM microscopy and for Ag–ONCs nanocomposites with (**d**) TEM, (**e**) SEM, and (**f**) AFM microscopy, respectively.

**Figure 4 materials-14-04198-f004:**
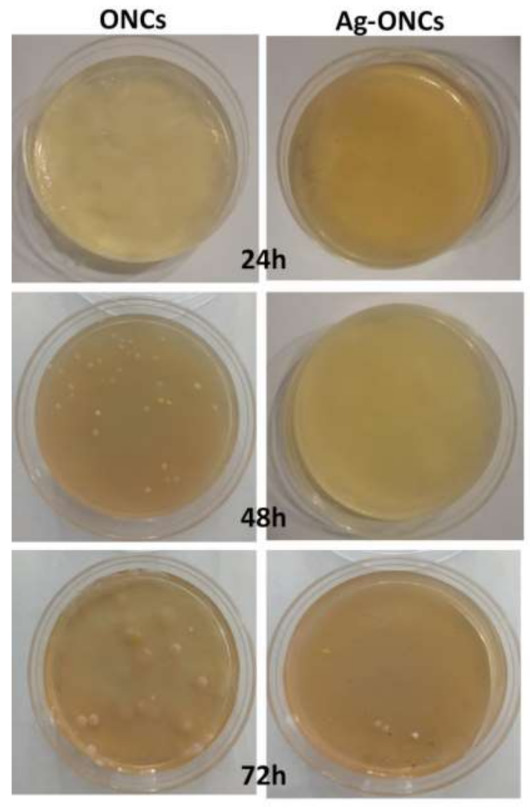
Comparison of antibacterial activity against *E. coli* of ONCs and Ag–ONCs nanocomposite after different incubation times.

**Figure 5 materials-14-04198-f005:**
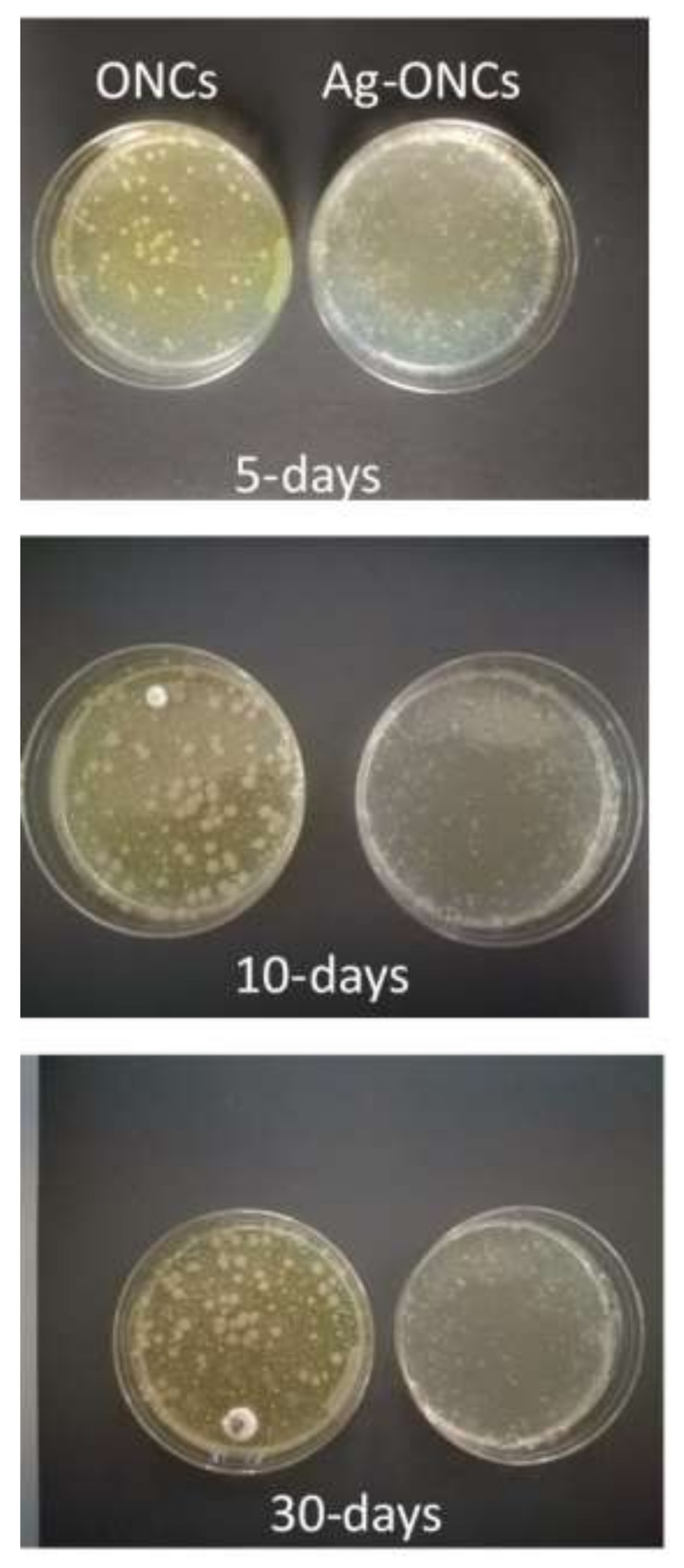
Comparison between ONCs and Ag–ONCs’ antibacterial properties after different incubation times. Petri dishes are stored at room temperature under natural light.

**Figure 6 materials-14-04198-f006:**
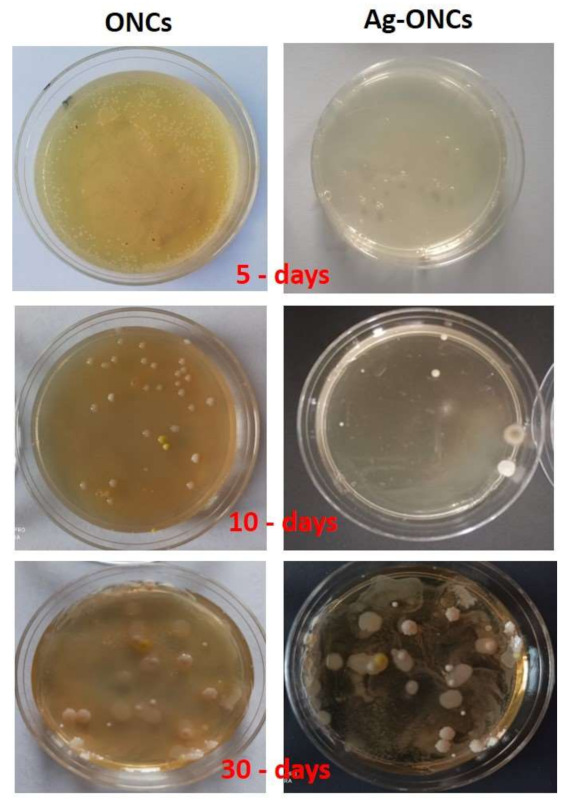
Comparison between ONCs and Ag–ONCs antibacterial properties after different incubation times. Petri dishes are stored in the dark to avoid UV light influence.

**Figure 7 materials-14-04198-f007:**
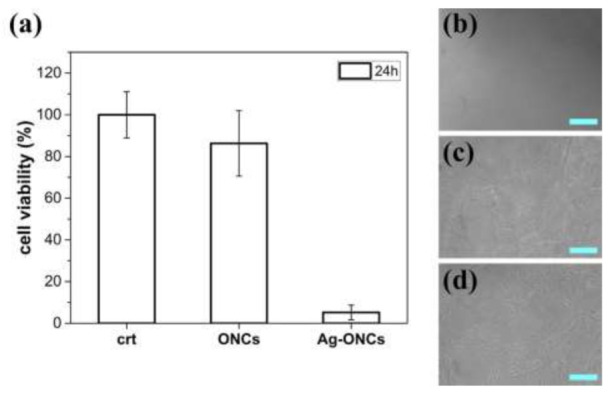
(**a**) Cell viability of NIH-3T3 cells after 24 h of exposure to Ag–ONCs or ONCs and without any NP treatment (crt). Data are presented as mean ± standard deviation (SD). Optical micrographs of NIH-3T3 cells treated with (**b**) Ag–ONCs or (**c**) ONCs and (**d**) without any NP treatment after 24 h of incubation (Scale bar = 100 µm).

**Table 1 materials-14-04198-t001:** XPS quantification and main peaks BE values of Ag–ONCs.

Name	Peak BE, eV	FWHM, eV	Atomic %	Bond
Ag3d5	368.2	2.0	0.6	Ag^0^
C1s—A	285.0	2.2	19.5	C–C, C–H
C1s—B	286.7	2.2	25.1	C–O
C1s—C	288.5	2.2	8.1	C=O, O–C–O
Na 1s	1072.0	2.3	3.0	Na^+^
O1s—A	531.8	2.6	12.5	C–O
O1s—B	533.1	2.6	30.8	C–O, O–C–O, H_2_O
S2p	169.1	2.9	0.4	Sulphite

## Data Availability

This study did not report any data.

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
