# Peer review of "Evaluation of Long–Lasting Antibacterial Properties and Cytotoxic Behavior of Functionalized Silver-Nanocellulose Composite"

_materials, 2021, doi:10.3390/ma14154198_

Round 1

Reviewer 1 Report

The authors propose a method for the formation of silver-nanocellulose composite and investigate its antibacterial and cytotoxic property. Indeed, some antibacterial and cytotoxic effects are confirmed by the respective studies presented by the authors. There is, however, no proof that all dissolved silver ions have transformed in nanoparticles. Also, during the nanocomposite preparation no procedure to remove remaining Ag+ ions was applied. For this reason, it remains doubtful if the antibacterial and cytotoxic behavior is due to the silver nanoparticles attached to the cellulose or to still remaining silver cations in the solution (esp. at high AgNO3 concentrations).

It should also be noted that all antibacterial and cytotoxic studies were conducted in aqueous environment. Such is hardly present for practical applications.

I would recommend publication after major revisions. It must, however, be clearly stated in the discussion and conclusions that no removal of silver ions during the nanocomposite preparation was applied and the observed effects could also (and most probably) originate from Ag+.  

Some additional corrections:

P5 – change peaks with reflections

P6 – change peaks with  absorption bands

P7 – reconsider “good homogeneity”. The values of about “40-95 nm in length and 5.94-10.42 nm in width” hardly point out to good homogeneity – the deviation is significant.

P9 – … with respect to…

P9 – … two sets …

Author Response

Response to referees:

We thank the reviewers for their careful reading of our manuscript and for their valuable questions and comments, that allowed us to improve its quality and consistency. We have revised the manuscript as a result of reviewers’ valuable feedbacks.

The changes in the revised version of the manuscript are made using the “Track Changes” function.

Please find below our point-to-point replies to the reviewers’ observations.

Reviewer 1

  1. The authors propose a method for the formation of silver-nanocellulose composite and investigate its antibacterial and cytotoxic property. Indeed, some antibacterial and cytotoxic effects are confirmed by the respective studies presented by the authors. There is, however, no proof that all dissolved silver ions have transformed in nanoparticles. Also, during the nanocomposite preparation no procedure to remove remaining Ag+ ions was applied. For this reason, it remains doubtful if the antibacterial and cytotoxic behaviour is due to the silver nanoparticles attached to the cellulose or to still remaining silver cations in the solution (esp. at high AgNO3 concentrations).
  2. Thanks to the reviewer for this good comment. The success of the formation of AgNp has been already proved in our previous works (see ref. 18 and ref. 20, in the revised version). According to the synthetic procedure, silver reduction has been carried out using a great excess of reducing/stabilizing agent (carboxylated nanocellulose) for an irradiation time (300 min) long enough to consider the reaction complete and quantitative (the UV/Vis spectrum did not change anymore after 240 min of UV irradiation). For this reason, we assume that the entire amount of silver precursor has been reduced to silver metallic. Furthermore, the other structural techniques confirmed that no signals related to the silver precursor (AgNO3) are revealed. However, we cannot exclude that very small amounts of Ag+ (not revealed by the instrumental resolutions) could be yet embedded in the colloidal solution and could partially contribute to enhance the antibacterial and cytotoxic Ag-ONCs properties. For clarity, some more explanations have been added in the synthetic procedure.

  1. It should also be noted that all antibacterial and cytotoxic studies were conducted in aqueous environment. Such is hardly present for practical applications.
  2. Thanks very much for referee’s observation. One of the aims of this work is to estabilish a green procedure for the fabrication of Ag nanocomposite. For this, water has been considered as the only solvent for the nanocomposite synthesis, in order to avoid toxic or no-environmental friendly solutions. Furthermore, at the best of our knowledge, preliminary antibacterial and cytotoxic studies, in lab scale, have usually conducted using water-based solutions. For these reasons, even if for practical applications different conditions could be used and considered, in our case, we limited the studies to the aqueous environment. Nevertheless, we also previously proved that the aqueous environment did not represent a great hindrance for practical applications, since, thanks to the characteristics of cellulose matrix, the nanocomposite can be easily embedded and stably fixed on porous substrates such as textile or paper, without losing their peculiarities (see ref.8, and ref.20 in the revised version).

  1. I would recommend publication after major revisions. It must, however, be clearly stated in the discussion and conclusions that no removal of silver ions during the nanocomposite preparation was applied and the observed effects could also (and most probably) originate from Ag+.  
  2. Thanks very much for referee’s comments. As pointed, more explanations about the synthetic procedure, the eventually presence of small amounts of Ag ions, and the effect on the observed final properties have been added in the text.

  1. Some additional corrections:

P5 – change peaks with reflections

P6 – change peaks with  absorption bands

P7 – reconsider “good homogeneity”. The values of about “40-95 nm in length and 5.94-10.42 nm in width” hardly point out to good homogeneity – the deviation is significant.

P9 – … with respect to…

P9 – … two sets …

R: Thanks very much for referee’s comments. We make the opportune corrections, according to the reviewer suggestions.

Reviewer 2 Report

The authors present the formation of silver nanoparticles onto an oxidised cellulose support for durable antibacterial activity. The synthesis of the nanoparticles is presented based on a previous route, the novelty relies on the fabrication of the antibacterial composite. Extensive characterisation was carried out to support the synthesis, which was good to see. I have the following concerns for the authors' attention:

In the introduction, the authors claim the use of silver as beneficial for health application; here the issue of environmental compatibility of silver should be discussed, especially since the authors claim the overall synthetic route to be green. 

In the introduction, it would also be beneficial to discuss other antibiotic free strategies for antibacterial activity, citing relevant papers such as Pharmaceutics 2020 (12) 711; ACS Appl. Bio Mater. 2019 (2) 4258.

The authors present cytotoxicity data against mouse embryonic fibroblasts. For health and food applications, such as the ones mentioned in this manuscript, the material should ideally display durable and selective antibacterial activity, so that toxic effects are only accomplished against bacteria but not against mammalian cells. It is not clear why cytotoxicity is a valuable properties with this regard. 

Re. the antibacterial activity, what is the exact mechanism for the antibacterial effect? Do silver nanoparticles leach out of the material with time so that silver acts against cells? This point is not clear and should be discussed.

Author Response

Response to referees:

We thank the reviewers for their careful reading of our manuscript and for their valuable questions and comments, that allowed us to improve its quality and consistency. We have revised the manuscript as a result of reviewers’ valuable feedbacks.

The changes in the revised version of the manuscript are made using the “Track Changes” function.

Please find below our point-to-point replies to the reviewers’ observations.

Reviewer 2

The authors present the formation of silver nanoparticles onto an oxidised cellulose support for durable antibacterial activity. The synthesis of the nanoparticles is presented based on a previous route, the novelty relies on the fabrication of the antibacterial composite. Extensive characterisation was carried out to support the synthesis, which was good to see. I have the following concerns for the authors' attention:

  1. In the introduction, the authors claim the use of silver as beneficial for health application; here the issue of environmental compatibility of silver should be discussed, especially since the authors claim the overall synthetic route to be green.

R.Thanks to the referee for this comment. It is well known that the environmental compatibility of a material is inversely proportional to its water solubility. If the water solubility is high, the probability that this material could be disperse in the environment is obviously higher. Furthermore, the ionic form of some metals makes them more environmental dangerous, respect to their corresponding nanoparticles structure. Another aspect that should be considered is the possibility to use the materials when embedded in a porous structure that can limit their diffusion in water/air/soil.

About the influence of silver nanoparticles, a very interesting review has been recently published, in which it is shown that the environmental impact of silver nanoparticles has to be discussed- (Health Impact of Silver Nanoparticles: A Review of the Biodistribution and Toxicity Following Various Routes of Exposure Zannatul Ferdous, Abderrahim Nemmar Int J Mol Sci. 2020 Apr; 21(7): 2375).

In our case, the synthesised AgNp-Nanocellulose nanocomposites show a very low solubility in water, and a homogeneous suspension can be obtained, in which AgNps are embedded in the polymeric matrix in a well stable way. These characteristics make the AgNp-ONCs not easily dispersible in the environment and porous substrates (as paper, textiles, sponge) that can be used to fabricate a device in which immobilize with the nanocomposites permits to avoid further dispersion of the active materials.

According to the referee’s suggestion, more details about the environmental impact of silver nanoparticles have been added in the introduction.

  1. In the introduction, it would also be beneficial to discuss other antibiotic free strategies for antibacterial activity, citing relevant papers such as Pharmaceutics 2020 (12) 711; ACS Appl. Bio Mater. 2019 (2) 4258.

R.I thank very much the referee for this suggestion. In the introduction, we added appropriate references about the use of silver nanoparticles in different antibiotic free strategies for antibacterial activities.

  1. The authors present cytotoxicity data against mouse embryonic fibroblasts. For health and food applications, such as the ones mentioned in this manuscript, the material should ideally display durable and selective antibacterial activity, so that toxic effects are only accomplished against bacteria but not against mammalian cells. It is not clear why cytotoxicity is a valuable properties with this regard.

R.Thanks to the reviewer for this good comment. Our aim in the work was to evaluate both long-lasting antibacterial properties and cytotoxicity of the Ag-ONCs, because Ag nanocomposites are of great interest especially in health and food fields, as we cited, but many others could be included, in which antibacterial properties OR cytotoxicity could be essential. In food packaging application or for the realization of specific medical devices (i.e. medical gauzes), specific (long-lasting) antibacterial properties are requested, while cytotoxicity towards cells is not obviously considered for them. On the other hand, since our Ag-ONCs show also very good cytotoxicity properties, potential use in biomedical applications could eventually be considered, even if a deep study of this specific application is far beyond our purposes. For this reason, the cytotoxicity of our Ag-ONC has also be evaluated, not only as a valuable property for the antibacterial behaviour, but also principally as a standing-alone property for further specific application in bio-medical field. Opportune modifications have been made in the text to avoid confusion and better clarify our aims.

  1. Re. the antibacterial activity, what is the exact mechanism for the antibacterial effect? Do silver nanoparticles leach out of the material with time so that silver acts against cells? This point is not clear and should be discussed.

R.Thanks to the referee for this interesting question. The antibacterial mechanism of the Ag-ONCs nanocomposites has been discussed in our previous works (see ref. 14 and ref. 16), in which the antibacterial behaviour of the AgNp-ONCs has been attributed to the synergic effect of the great amounts of different functionalities on the cellulosic surface, combined with the natural properties of Ag. This combination permits the AgNPs to better interact with the cell surface of different bacteria. This mechanism has been already reported in other studies (Ruparelia JP, Chatterjee AK, Duttagupta SP, Mukherji S (2008) Strain specificity in antimicrobial activity of silver and copper nanoparticles. Acta Biomater 4:707–716. https:// doi.org/10.1016/j.actbio.2007.11.006 – ref. 47 in the revised version of the manuscript). Furthermore, according to Durán (N. Durán, M. Durán, M.B. de Jesus, A.B. Seabra, W.J. Fávaro, G. Nakazato, Silver nanoparticles: A new view on mechanistic aspects on antimicrobial activity, Nanomedicine Silver nanoparticles: A new view on mechanistic aspects on antimicrobial activity, Nanomedicine Nanotechnology, Biol. Med. 12 (2016) 789–799. https://doi.org/10.1016/j.nano.2015.11.016 - ref. 48 in the revised version of the manuscript), the antimicrobial behaviour of AgNPs-ONC bio-nanocomposites can be explained by considering the release of AgNPs, which can interact with sulfur-containing proteins in the cell membrane thus altering their permeability and leading to cell leakage. Furthermore, AgNPs can also interfere and destroy the normal metabolism of cells, infiltrating their inner shell.

A more deep and clear discussion about the antibacterial behaviour of Ag-ONCs has been added in the text and new references have been included, supporting our comments.

Round 2

Reviewer 1 Report

In my opinion the introduced changes in the text have improved the manuscript. I recommend publication in this form.

Reviewer 2 Report

The authors have di substantially addressed my previous concerns and the manuscript can be approved for publication.